# Genome-wide meta-analysis of muscle weakness identifies 15 susceptibility loci in older men and women

Garan Jones [1,58], Katerina Trajanoska [2,3,58], Adam J. Santanasto [4,58], Najada Stringa [5,58], Chia-Ling Kuo[6,58], Janice L. Atkins [1,58], Joshua R. Lewis [7,8,9], ThuyVy Duong[10], Shengjun Hong[11], Mary L. Biggs[12], Jian'an Luan [13], Chloe Sarnowski [14], Kathryn L. Lunetta [14], Toshiko Tanaka[15], Mary K. Wojczynski[16], Ryan Cvejkus [4], Maria Nethander [17,18], Sahar Ghasemi[19,20], Jingyun Yang [21], M. Carola Zillikens[2], Stefan Walter [22,23], Kamil Sicinski[24], Erika Kague [25], Cheryl L. Ackert-Bicknell[26], Dan E. Arking[10], B. Gwen Windham[27], Eric Boerwinkle[28,29], Megan L. Grove[28], Misa Graff [30], Dominik Spira[31,32], Ilja Demuth[31,32,33], Nathalie van der Velde[34], Lisette C. P. G. M. de Groot [35], Bruce M. Psaty[36,37], Michelle C. Odden[38], Alison E. Fohner[39], Claudia Langenberg [13], Nicholas J. Wareham [13], Stefania Bandinelli[40], Natasja M. van Schoor[5], Martijn Huisman[5], Qihua Tan [41], Joseph Zmuda[4], Dan Mellström[17,42], Magnus Karlsson[43], David A. Bennett[21], Aron S. Buchman [21], Philip L. De Jager [44,45], Andre G. Uitterlinden [2], Uwe Völker [46], Thomas Kocher [47], Alexander Teumer [20], Leocadio Rodríguez-Mañas[48,23], Francisco J. García[49,23], José A. Carnicero[23], Pamela Herd[50], Lars Bertram [11], Claes Ohlsson[17,51], Joanne M. Murabito[52], David Melzer [1], George A. Kuchel [53,59], Luigi Ferrucci [54,59], David Karasik [55,56,59], Fernando Rivadeneira [2,3,59], Douglas P. Kiel [57,59] & Luke C. Pilling [1,59✉]

Low muscle strength is an important heritable indicator of poor health linked to morbidity and mortality in older people. In a genome-wide association study meta-analysis of 256,523 Europeans aged 60 years and over from 22 cohorts we identify 15 loci associated with muscle weakness (European Working Group on Sarcopenia in Older People definition: $n = 48,596$ cases, 18.9% of total), including 12 loci not implicated in previous analyses of continuous measures of grip strength. Loci include genes reportedly involved in autoimmune disease (*HLA-DQA1* $p = 4 \times 10^{-17}$), arthritis (*GDF5* $p = 4 \times 10^{-13}$), cell cycle control and cancer protection, regulation of transcription, and others involved in the development and maintenance of the musculoskeletal system. Using Mendelian randomization we report possible overlapping causal pathways, including diabetes susceptibility, haematological parameters, and the immune system. We conclude that muscle weakness in older adults has distinct mechanisms from continuous strength, including several pathways considered to be hallmarks of ageing.

A list of author affiliations appears at the end of the paper.

Age-associated loss of muscle strength (termed dynapenia)[1] is one of the characteristic changes occurring with advancing age, and muscle weakness is considered a fundamental component of frailty and sarcopenia[2]. Individuals over 70 years old typically demonstrate up to 20% lost muscle mass compared with individual in their twenties[3]. Although definitions of reduced muscle function in older people have focused on loss of muscle mass (sarcopenia) evidence now shows that muscle weakness itself is often more predictive of negative health outcomes[4]. Muscle weakness causes difficulties in daily functioning (i.e., disability) and low muscle strength (measured as hand grip strength, considered a biomarker of general dynapenia) is predictive of future morbidity and mortality[3] over the long term[5]. Despite intensive research, causes of and contributors to muscle weakness in later life remain to be fully elucidated[6]. Importantly, muscle strength is heritable (48–55% in 1757 male twin pairs aged 45–96)[7], and can thus be used for genetic investigations.

Previously a genome-wide association study (GWAS) by the CHARGE (Cohorts for Heart and Aging Research in Genomic Epidemiology) consortium identified two loci associated with maximum hand grip strength (as a quantitative trait) in 27,581 Europeans aged 65 and over[8]. Another study on maximum hand grip strength (divided by weight) in mostly middle-aged UK Biobank participants (334,925 people aged 40–70, mean aged 56) identified and replicated 64 loci, many of which are known to have a role in determining anthropometric measures of body size[9,10]. These previous studies that considered grip strength as a continuous phenotype across young and old individuals may not provide insights into the age related loss of muscle strength that leads to a magnitude of weakness sufficient to call it a disease. Given the limited data on genetic contributions to a clinically meaningful level of muscle weakness in older adults, we aimed to determine the genetic variants and investigate causal pathways associated with low measured grip strength.

In this work, we report a GWAS of weakness in 256,523 older adults (aged 60+ years) of European ancestries from the CHARGE consortium. The primary analysis was based on the established 2010 European Working Group on Sarcopenia in Older People (EWGSOP) definition of low grip strength, and results were compared to an analysis of the alternative Foundations of the National Institutes of Health (FNIH) definition based on its association with functional outcomes, with additional analyses stratified by sex. The associated loci and subsequent pathway and Mendelian randomization analysis reveal causal pathways to weakness at older ages distinct from overall strength during the life course, highlighting specific diseases (such as osteoarthritis) and link to hallmark aging mechanisms such as cell cycle control.

## Results

**Study description.** The meta-analysis comprised 256,523 individuals of European descent aged 60 years or older at assessment from 22 independent cohorts with maximum hand grip strength recorded—including the UK Biobank, the US Health and Retirement Study, the Framingham Heart Study, and others. In total, 46,596 (18.9%) of all participants had muscle weakness (dynapenia) based on hand grip strength (EWGSOP definition: grip strength <30 kg Male; <20 kg Female). Individual study characteristics are described in the Supplementary Information and in Supplementary Table 1.

Our primary analysis of EWGSOP definition low grip strength will be described first, with subsequent additional analyses described in later sections: these include analysis of males and females separately and use of the alternative low grip strength criteria provided by the FNIH data.

**GWAS of low muscle strength identifies 15 loci.** We found 15 genomic risk loci to be associated ($p < 5 \times 10^{-8}$; 8 loci $p < 5 \times 10^{-9}$) with EWGSOP definition low hand grip strength in our GWAS meta-analysis of 22 cohorts ($n = 256,523$, $n = 48,596$ cases), adjusted for age, sex, and technical covariates (Fig. 1, Table 1; Supplementary Data 1). The strongest associations were with variants close to *HLA-DQA1* (rs34415150, beta/log-OR per G allele = 0.0833, $p = 4.4 \times 10^{-17}$), *GDF5* (rs143384, beta per A allele=0.0545, $p = 4.5 \times 10^{-13}$) and *DYM* (rs62102286, beta per T allele=0.0487, $p = 5.5 \times 10^{-11}$). Twelve of the fifteen lead SNPs from the GWAS have not previously been identified in studies of grip strength analyzed on a continuous scale across all ages (Supplementary Data 2 & 3) and only 3 of the 64 loci associated with overall muscle strength[11] are significant in our analysis of low strength (Supplementary Data 4). This included the three most strongly associated variants near *HLA-DQA1* (previously implicated in rheumatoid arthritis: see Supplementary Data 2), *GDF5* ('Growth differentiation factor 5': previously implicated in height, waist hip ratio, muscle mass, and osteoarthritis) and *DYM* ('Dymeclin': implicated in in height). Six other variants were previously linked to height and four to osteoarthritis. None were significantly ($p < 5 \times 10^{-8}$) associated with lean muscle mass, although rs10952289 near AOC1 is nominally associated with appendicular lean muscle mass ($p = 6 \times 10^{-4}$)[12]. The test of cohort heterogeneity in METAL for all 15 lead SNPs was not statistically significant (nominal het $p > 0.05$). Full summary statistics for the meta-analysis are available for download (visit the Musculoskeletal Knowledge Portal http://www.mskkp.org/ or the GWAS catalogue https://www.ebi.ac.uk/gwas/).

Overall, two of the fifteen identified lead variants (or proxies) have not previously been implicated in anthropometric or musculoskeletal phenotypes in the GWAS catalogue (see Supplementary Data 2). This included *ALDH1A2* ('Aldehyde Dehydrogenase 1 Family Member A2': involved in the synthesis of retinoic acid), and a variant near *FTO* ('FTO Alpha-Ketoglutarate Dependent Dioxygenase': involved in the oxidative demethylation of different forms of RNA). Although the lead *ALDH1A2* SNP itself has not been identified in previous GWAS, other independent variants ($R^2 < 0.6$) at the same locus (e.g., rs3204689) have been found to be associated with osteoarthritis (Supplementary Data 2).

The Lambda GC (genomic control, $\lambda_{GC}$) value was high (1.13; see Supplementary Fig. 1 for QQ plot), however the intercept from Linkage Disequilibrium Score Regression (LDSC) analysis was close to 1 (0.97, SE 0.007), indicating that the inflation in test statistics is primarily due to polygenicity (many variants with small effects on low grip strength), rather than bias due to population stratification[13]. An intercept below 1 is not unusual for analyses adjusted with genomic control. The single nucleotide polymorphism (SNP) based heritability ($h^2$) of low grip strength was 0.044 (SE 0.0027), i.e., 4.4%, by LD Score Regression.

In sex-stratified analysis there were eight significant genomic risk loci associated with EWGSOP low grip strength in females only (total $n = 132,443$ with $n = 33,548$ cases, 25.3%; see Supplementary Table 2). Seven of the eight loci were either present in the main analysis or were correlated with corresponding variants, however rs7185040 (chr16: 2145787), mapped to gene *PKD1*, was only significant in the analysis of females, although the association is borderline in the analysis of males and females together (females $p = 3 \times 10^{-8}$; combined analysis $p = 5.5 \times 10^{-8}$).

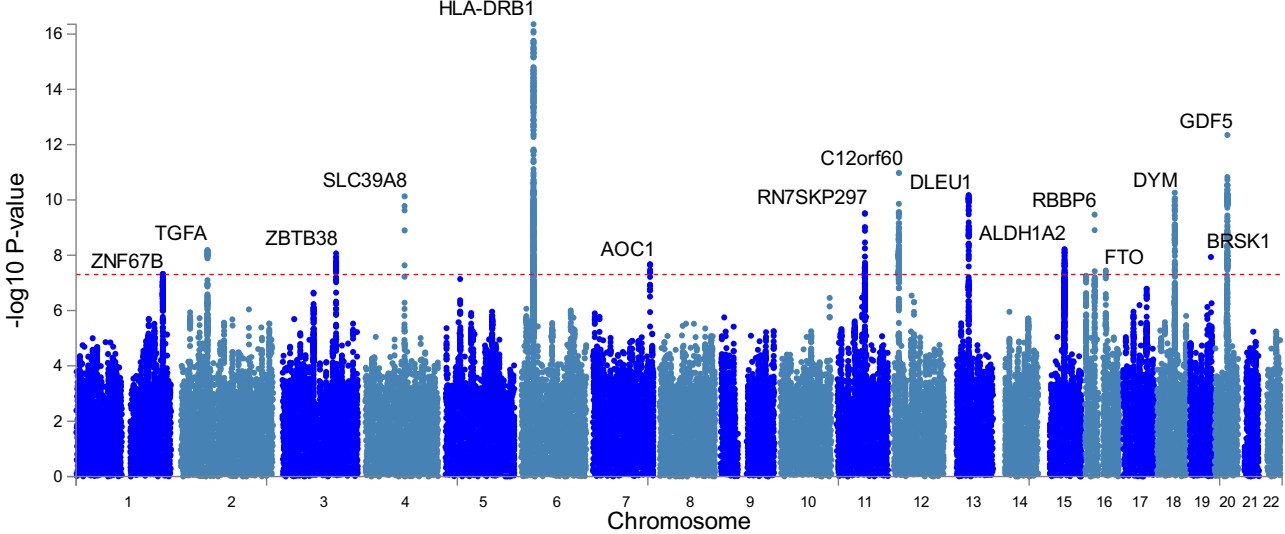

**Fig. 1 Manhattan plot of low grip strength genome-wide association study -log10 p values.** The p values are from a fixed-effects meta-analysis of 256,523 Europeans aged 60 or older from 22 cohorts. The outcome was low hand grip strength grip strength cutoff (males <30 Kg, females <20 Kg). The x-axis is the chromosomal location, and the y-axis is the $-\log_{10} p$ value for each genetic variant. The horizontal red line is the threshold for genome-wide significance ($p < 5 \times 10^{-8}$). Fifteen genomic loci cross the threshold, and the lead variant (most significantly associated with low strength) is described in Table 1. The nearest gene is displayed for each locus.

**Table 1 Fifteen genomic risk loci associated with low grip strength in 256,523 older men and women.**

| RSID | Chr | BP (b37) | EA | OA | EAF | OR | p value | Nearest gene | GTEx increased | GTEx decreased |
|---|---|---|---|---|---|---|---|---|---|---|
| rs34415150 | 6 | 32560477 | G | A | 0.18 | 1.087 | $4.4 \times 10^{-17}$ | *HLA-DRB1* | *HLA-DQA2; HLA-DRB6; HLA-DQB2; HLA-DOB* | *HLA-DQA1; HLA-DRB1; HLA-DQB1; HLA-DQB1-AS1* |
| rs143384 | 20 | 34025756 | A | G | 0.59 | 1.056 | $4.5 \times 10^{-13}$ | *GDF5* | *UQCC1; FAM83C; CPNE1* | *GDF5; RPL36P4* |
| rs62102286 | 18 | 46592408 | T | G | 0.56 | 1.050 | $5.5 \times 10^{-11}$ | *DYM* | | *DYM* |
| rs3118903 | 13 | 51099577 | A | G | 0.22 | 1.059 | $6.7 \times 10^{-11}$ | *DLEU1* | | *RNASEH2B-AS1* |
| rs13107325 | 4 | 103188709 | T | C | 0.07 | 1.094 | $7.4 \times 10^{-11}$ | *SLC39A8* | | *UBE2D3* |
| rs11236213 | 11 | 74394369 | G | A | 0.69 | 1.052 | $3.0 \times 10^{-10}$ | *RN7SKP297* | *KCNE3; POLD3* | *RP11-864N7.4; CHRDL2* |
| rs34464763 | 12 | 15032860 | A | T | 0.39 | 1.056 | $3.2 \times 10^{-10}$ | *C12orf60* | RP11-233G1.4 | *ERP27; SMCO3; C12orf60; MGP* |
| rs143459567 | 16 | 24600412 | T | C | 0.04 | 1.126 | $3.4 \times 10^{-10}$ | *RBBP6* | | |
| **rs2899611** | **15** | **58327347** | **G** | **T** | **0.50** | **1.044** | **$6.0 \times 10^{-9}$** | ***ALDH1A2*** | | ***ALDH1A2*** |
| rs958685 | 2 | 70703847 | C | A | 0.49 | 1.044 | $6.5 \times 10^{-9}$ | *TGFA* | *TGFA* (Ts)[a] | *TGFA* (Bcor, Bcau, Bhyp, Bacc)[a] |
| rs7624084 | 3 | 141093285 | T | C | 0.56 | 1.044 | $8.5 \times 10^{-9}$ | *ZBTB38* | *ZBTB38* (Skse, Esom, Snse)[a] | *ZBTB38* (Wb, Thy, Adips, Ts)[a] |
| rs79723785 | 19 | 55818225 | C | T | 0.02 | 1.182 | $1.2 \times 10^{-8}$ | *BRSK1* | *HSPBP1** | |
| rs10952289 | 7 | 150524681 | T | C | 0.66 | 1.044 | $2.1 \times 10^{-8}$ | *AOC1* | *AOC1* (Haa, Hlv, Liv); *TMEM176B* (Haa)[a] | *AOC1* (Esom, Adipv, Thy, Esog)[a] |
| **rs8061064** | **16** | **53912364** | **A** | **T** | **0.46** | **1.042** | **$3.6 \times 10^{-8}$** | ***FTO*** | | |
| rs12140813 | 1 | 227776827 | T | C | 0.19 | 1.052 | $4.8 \times 10^{-8}$ | *ZNF678* | *JMJD4* | *SNAP47* |

p value = fixed-effects meta-analysis p value; Nearest gene = on GRCh37, gene names italicized; GTEx increased/decreased = Top four genes with known expression associations with the lead SNP in GTEx v8, ordered by p value. Rows in bold are those not found in previously published GWAS (see Supplementary Data 2 and 3).
Chr chromosome, BP base pair position, genome build 37, EA effect allele, OA other allele, EAF effect allele frequency, OR odds ratio of having low grip strength (EWGSOP criteria) per allele.
[a]GTEx v8 differential expression by tissue – Ts = Testis; Bcor = Brain – Cortex; Bcau = Brain - Caudate (basal ganglia); Bhyp = Brain – Hypothalamus; Bacc = Brain – Anterior cingulate cortex; Asub = Adipose – subcutaneous; Wb = Whole blood; Esom = Esophagus – mucosa; Skse = Skin sun exposed lower leg; Snse = Skin non-sun exposed lower leg; Thy = Thyroid; Adips = Adipose – Subcutaneous; Adipv = Adipose - Visceral (Omentum); Esog = Esophagus - Gastroesophageal Junction; Haa = Heart - Atrial Appendage; Hlv = Heart - Left Ventricle; Liv = Liver. See Supplementary Data 1 for METAL output, and Supplementary Data 6 for GTEx

The analysis of males only (total $N = 118,371$ with 13,327 cases, 11.3%) identified three genomic loci associated with the EWGSOP low grip strength definition. Two of these variants appeared to be distinct signals from the overall analysis and were not associated with low grip in females (see Supplementary Table 3 for details): rs774787160 mapped to gene *DSCAM* (males $p = 1 \times 10^{-8}$; females $p = 0.9$) and rs145933237 mapped to mir-466, which was only nominally associated in females (males $p = 2 \times 10^{-8}$; females $p = 0.01$).

In the analysis of 116 mitochondrial genetic variants (MAF >0.01) available in the UK Biobank directly genotyped microarray data, no variants reached "genome-wide" significance ($p > 5 \times 10^{-8}$). Two were associated with EWGSOP-defined low hand grip strength at nominal significance ($p < 0.00043$, i.e., Bonferroni-adjustment for

mitochondrial variants). rs41518645 is a missense variant (p.Asp171Asn) in *MT-CYB*, identified in Plink logistic regression analysis ($p = 0.0003$). rs201950015 is intronic, located between genes *CO1* and *ATP6/8* ($p = 0.00042$). These findings need further scrutiny in studies assessing the influence of mitochondrial dysfunction on muscle function and metabolism. See Supplementary Data 5.

**GWAS of low grip strength based on FNIH criteria.** In secondary analysis we performed GWAS using the low grip strength definition published by the FNIH[14]. This criterion uses lower grip strength cut-offs (<26 kg for males and <16 kg for females) than the EWGSOP definition[15], resulting in fewer cases ($n = 19,345$, 7.6% of total). Five loci were significant in the analysis ($p < 5 \times 10^{-8}$), only

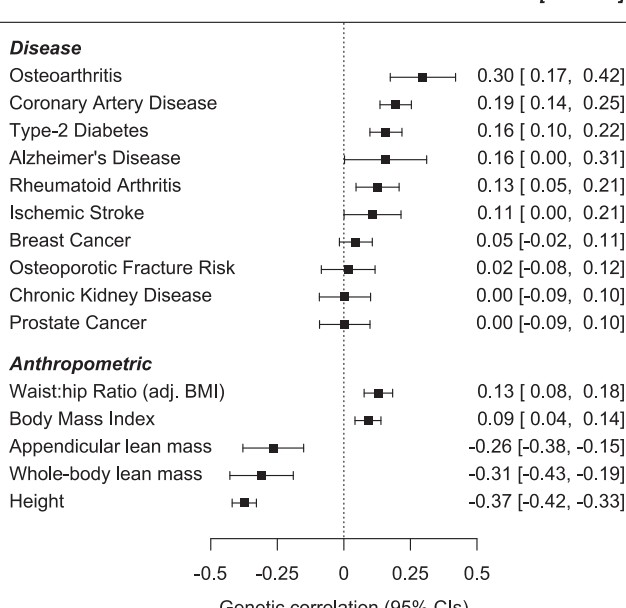

| Genetic Correlation | rG [95% CIs] |
|---|---|
| **Disease** | |
| Osteoarthritis | 0.30 [ 0.17, 0.42] |
| Coronary Artery Disease | 0.19 [ 0.14, 0.25] |
| Type-2 Diabetes | 0.16 [ 0.10, 0.22] |
| Alzheimer's Disease | 0.16 [ 0.00, 0.31] |
| Rheumatoid Arthritis | 0.13 [ 0.05, 0.21] |
| Ischemic Stroke | 0.11 [ 0.00, 0.21] |
| Breast Cancer | 0.05 [-0.02, 0.11] |
| Osteoporotic Fracture Risk | 0.02 [-0.08, 0.12] |
| Chronic Kidney Disease | 0.00 [-0.09, 0.10] |
| Prostate Cancer | 0.00 [-0.09, 0.10] |
| **Anthropometric** | |
| Waist:hip Ratio (adj. BMI) | 0.13 [ 0.08, 0.18] |
| Body Mass Index | 0.09 [ 0.04, 0.14] |
| Appendicular lean mass | -0.26 [-0.38, -0.15] |
| Whole-body lean mass | -0.31 [-0.43, -0.19] |
| Height | -0.37 [-0.42, -0.33] |

Genetic correlation (95% CIs)

**Fig. 2 Low grip strength genetic correlations with ten common diseases and five anthropometric traits.** Genome-wide genetic correlations between low muscle strength and published summary statistics for common age-related diseases and low muscle strength risk factors. N = meta-analysis of 256,523 Europeans (biologically independent samples) aged 60 or older from 22 cohorts. Data presented are genetic correlation (rG) ± 95% Confidence Intervals from LD-Score Regression analysis of 1.2 million SNPs. Full results available in Supplementary Table 5.

one of which was not identified in the EWGSOP low grip strength analysis described previously (see Supplementary Table 4, either the same SNP or in high LD with the EWGSOP lead SNP at that loci, for example rs3771501 and rs958685 $R^2 = 0.90$). This single base-pair deletion (rs1403785912–chr9:4284961:T:-) mapped to *GLIS3* ("GLIS family zinc finger 3"—a repressor and activator of transcription), and may be specifically associated with more strict definitions of weakness (EWGSOP $p$ value $= 1.2 \times 10^{-3}$).

**Gene expression and pathways.** We used data from the Genotype-Tissue Expression project (GTEx) v8 to identify whether the variants associated with low grip strength affect expression of genes (Table 1; Supplementary Data 6). Of the top 15 EWGSOP-associated variants 12 are eQTLs for at least one gene. For 8 of these, the nearest gene to the variant by chromosomal location is known to have altered expression, but other genes in the locus may also be affected. This is consistent with a recent study showing that the "nearest gene" is often a good candidate for being a causal pathway[16]. For the top two loci (*HLA-DQA1* and *GDF5*) the variants are eQTLs for these nearest genes, however for the *SLC39A8* locus the lead SNP (rs13107325) is not an eQTL for *SLC39A8*, but is an eQTL for *UBE2D3* ('Ubiquitin Conjugating Enzyme E2 D3') in the aorta.

In MAGMA analysis we found 80 GO processes enriched in low grip strength-associated genes (see Supplementary Data 7 for details), mainly involved in the immune system and antigen presentation.

Using MetaXcan[17] we identified 24 genes with expression in skeletal muscle significantly enriched in the low grip strength GWAS, after Benjamini-Hochberg adjustment for multiple testing (Supplementary Data 8). We used the International Mouse Phenotyping Consortium database (www.mousephenotype.org)

to investigate the possible phenotypes associated with the genes highlighted by MetaXcan, with many having clear effects on relevant phenotypes such as "growth", "lean mass", "body weight", "angiogenesis", "thymus involution", and "lipid metabolism" (Supplementary Data 9).

We used LDSC-SEG to determine tissue-specific gene expression and chromatin modification enrichment in the low grip strength GWAS results[18]. We found no significant enrichment for the genetic determinants of low grip strength in expression profiles and epigenetic changes after adjustment for multiple testing (Benjamini-Hochberg-adjusted false discovery rate >0.05). See Supplementary Data 10 for details.

**Genetic correlations and Mendelian randomization.** We assessed ten common age-related diseases for their genetic correlation with low grip strength (Fig. 2) using published genome-wide summary statistics and LDSC[13]. The largest genetic overlap was with osteoarthritis (29.7% genetic correlation, SE 6.3%), but also strong positive correlations with coronary artery disease (19.5%, SE 3.0%), type-2 diabetes (15.8%, SE 3.1%) and rheumatoid arthritis (12.7%, SE 7.9%). We observed no significant genetic correlation after multiple-testing correction with the remaining diseases examined (osteoporotic fracture risk, Alzheimer's disease, stroke, chronic kidney disease, breast cancer and colorectal cancer). We also determined genetic correlations with five anthropometric traits (Fig. 2), and found significant positive correlations with waist:hip ratio (13.0%, SE 2.7%) and BMI (9.1%, SE 2.5%), i.e. greater adiposity correlated with weakness in 60+ year olds. Significant negative correlations were observed with lean muscle mass (whole body: −30.9%, SE 6.1%; and appendicular: −26.5%, SE 5.8%) and with height (−37.4%, SE 2.3%). See Supplementary Table 5 for details.

We examined 83 traits in Mendelian randomization analysis to find evidence for shared causal pathways with weakness (low grip EWGSOP) at older ages: primary results presented are betas from inverse variance-weighted regression using the "TwoSampleMR" R package[19] (Fig. 3; Supplementary Data 11). We found significantly increased likelihood of weakness (multiple testing-adjusted $p$ values < 0.05) with genetically predicted rheumatoid arthritis (Odds ratio = 1.03, Benjamini-Hochberg adjusted $p$ value $= 5.3 \times 10^{-4}$), presence of type-2 diabetes (OR = 1.05, BH $p = 2.5 \times 10^{-3}$), or asthma and allergic disease (OR = 1.07, BH $p = 9.4 \times 10^{-3}$) (Supplementary Data 11). Genetic predisposition to greater age of menarche (OR = 0.92, BH $p = 5.3 \times 10^{-4}$), birth weight (OR = 0.80, BH $p = 5.3 \times 10^{-4}$) and waist-hip ratio (WHR) adjusted for BMI in women only (OR = 0.85, BH $p = 5.3 \times 10^{-4}$) were protective of low grip strength as defined by the EWGSOP definition in both sexes (after adjustment for multiple testing). For each significant analysis we also examined the results from the weighted-median and MR-Egger tests to check consistency and for horizontal pleiotropy. Only birth weight had an MR-Egger beta that was inconsistent with the main effect (−0.03 compared to −0.2), although the intercept did not significantly deviate from 0 ($p = 0.1$) and the MR-Egger confidence intervals overlap the IVW effect (95% CIs −0.33 to 0.27). In addition, the WHR association should be interpreted with caution, as the analysis of WHR variants associated with both sexes were not statistically significant (nominal IVW $p = 0.01$).

The analysis of females only also identified depression (OR = 1.21, BH $p = 2.75 \times 10^{-2}$), and colorectal cancer (OR = 1.06, BH $p = 2.75 \times 10^{-2}$) (Supplementary Data 12). Although the MR-Egger intercept for depression was not statistically different from 0, there is potential for horizontal pleiotropy confounding this result (seen on Supplementary Fig. 3).

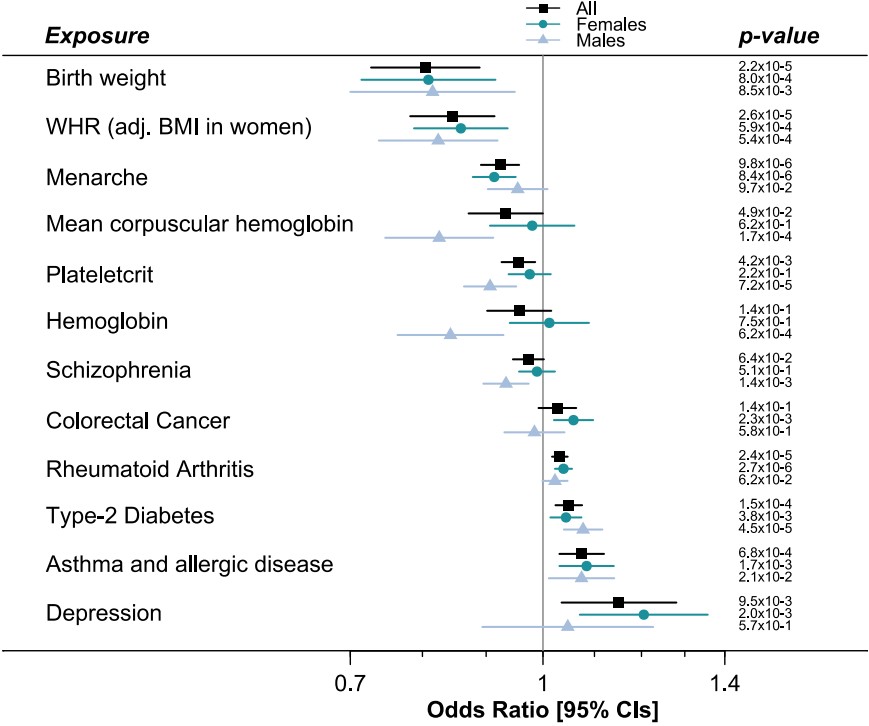

**Fig. 3 Traits sharing causal pathways with low grip strength at older ages identified in Mendelian Randomization analysis.** In Mendelian randomization analyses we estimated whether 83 exposures may share causal pathways with low grip strength in people aged 60 and older. Those identified as significant (multiple testing-adjusted $p < 0.05$) in at least one analysis (all participants, or males or females separately) are included in the figure. *WHR (adj. BMI in Women) = Waist-Hip Ratio SNPs identified in GWAS analysis of females only, adjusted for Body Mass Index. N = meta-analysis of 256,523 Europeans (biologically independent samples) aged 60 or older from 22 cohorts. Data presented as Odds Ratios +/− 95% Confidence Intervals. Unadjusted p value (two-sided) from IVW regression analysis of exposure SNPs effect on low grip strength (EWGSOP definition). Full results of Mendelian randomization available in Supplementary Data 11, 12 and 13.

In the males, type-2 diabetes significantly increased odds of EWGSOP low grip (OR = 1.08, BH $p = 3.0 \times 10^{-3}$) whereas greater plateletcrit (the volume occupied by platelets in the blood as a percentage) (OR = 0.91, $p = 3.0 \times 10^{-3}$) and other haematological parameters appear to be protective (Supplementary Data 13).

To explore the effect of genetic predisposition to low grip strength at older ages we created an unweighted genetic risk score (GRS) in the UK Biobank European sample by summing the number of low grip strength-associated alleles (15 genetic variants so 30 alleles, mean number of alleles = 12.6, SD = 2.3). We opted to use an unweighted score as use of weights from analyses including the discovery sample can bias associations and lead to overestimated effects (so-called "winner's curse")[20]. We first confirmed the association with low grip strength in UK Biobank participants (OR per allele 1.036: 95% CIs 1.030 to 1.041, $p = 6 \times 10^{-40}$). The low grip GRS was also associated with increased Frailty Index[21] (increase in points per allele = 0.013: 0.006 to 0.021, $p = 4 \times 10^{-4}$).

**Low grip strength loci independence from musculoskeletal traits and diseases**. To determine whether the genetic variants associated with low grip strength identified in the GWAS were independent of anthropometric traits or prevalent musculoskeletal comorbidities we performed regression analyses in the UK Biobank cohort with adjustment for the following covariates: height, weight, skeletal muscle mass (determined using bioimpedance analysis), osteoarthritis, Rheumatoid arthritis, osteoporosis, Dupuytren's contracture (one or more fingers permanently bent), and rhizarthrosis (arthritis of the thumb). Disease diagnoses were either self-reported, hospital diagnosed,

or inferred from relevant surgical procedures (for example Palmar Fasciectomy to treat Dupuytren's contracture), and hip or knee replacements resulting from osteoarthritis: UK Biobank hospital episode statistics diagnosis and operations data available up to March 2017. See Supplementary Table 6 for diagnostic and surgical codes used.

The association between 8 of the 15 EWGSOP loci and low grip strength was attenuated after adjusting for height, including rs143384 (initial UKB $p = 3.7 \times 10^{-11}$; adjusted UKB $p = 3.8 \times 10^{-2}$) and rs7624084 (initial UKB $p = 9.3 \times 10^{-7}$; adjusted UKB $p = 4.9 \times 10^{-1}$). Adjustment for weight or BMI did not substantially attenuate any of the associations. Overall, the associations were not attenuated by adjustment for osteoarthritis, Rheumatoid arthritis, osteoporosis, Dupuytren's, or rhizarthrosis. See Supplementary Data 14 for detailed results.

## Discussion

In this study of 256,523 Europeans aged 60 years and over we found that 15 genetic loci were associated with the EWGSOP definition of low grip strength (dynapenia), plus two additional loci for the FNIH definition that used a more strict definition for muscle weakness. Only three of these are known to be associated with continuous strength measures in GWAS, and only 3 of the 64 known overall strength loci are associated with clinically low grip strength used in our study. These suggest that the genetic causes of clinically meaningful weakness at older ages are partly distinct. Two of the low grip strength-associated genetic signals identified have not been reported by GWAS prior to the time of analysis (March 2020), further demonstrating that low strength in older people may have distinct genetic underpinnings. We did also find prominent overlaps with osteoarthritis and Rheumatoid

arthritis, and also with cardiovascular disease and type-2 diabetes. Additional links to asthma and allergy were also found. The pathways implicated appear to include hallmark mechanisms of ageing[22], for example cell cycle control related to the cancer control retinoblastoma pathway. However other ageing pathways such as telomere length[23], and many lifespan-associated loci including APOE[24], were not associated.

The strongest association found was rs34415150, near the HLA-DQA1 gene. Genetic variants at this locus have been implicated in a wide range of conditions, including autoimmune diseases such as Rheumatoid arthritis[25], and continuous grip strength[9]. HLA haplotypes HLA-DQA1*03:01 and HLA-DRB1*04:01, have been previously linked to sarcopenia in older UK Biobank participants[26]. HLA-DQA1 is associated with chronic inflammation in muscle of untreated children with juvenile dermatomyositis (inflammatory myopathies in children, which one of the characteristics is muscle weakness)[27]. In addition, in a multi-trait analysis of age-related diseases HLA-DQA1 was identified and it may therefore contribute to underlying ageing mechanisms as a "geroscience locus"[28].

Overall 6 of the 15 genomic risk loci for EWGSOP low grip strength have been previously associated (or are in LD) with osteoarthritis (rs143384 – GDF5, rs13107325 – SLC39A8, rs34464763 – C12orf60, rs2899611 - ALDH1A2, rs958685 – TGFA and rs79723785 - BRSK1), of which two are also linked to adiposity, a known risk factor for osteoarthritis[29]. We found that rs143384 in the 5' untranslated region of growth/differentiation factor 5 (GDF5) was the second most strongly associated variant with low grip strength. GDF5 is a protein in the transforming growth factor beta (TGF-β) family, with key roles in bone and joint development[30,31]. It was the first locus identified for osteoarthritis[32], with a reported odds ratio of 1.79, as well as one of the first identified for height[33]. GDF5 is known to reduce expression of cartilage extracellular matrix-degrading enzymes in human primary chondrocytes[34], thereby may be an potential intervention target for avoiding weakness at older ages, although work is needed to determine when intervention would be most effective. In follow-up analyses we found that the association between rs143384 and low grip strength was independent of prevalent osteoarthritis in UK Biobank participants, although we cannot rule out an effect of sub-clinical osteoarthritis. However, the association was completely attenuated after adjustment for standing height, suggesting the effect of the variant is mediated by developmental traits such as bone length. Mice with loss of Gdf5 function exhibited severely impaired knee development, and the regulatory region pinpointed to mediate this effect in humans includes osteoarthritis-associated genetic variants[35]. Although the observed association between variants mapping to GDF5 and low strength might be mediated by hand osteoarthritis pain compromising grip strength, there is evidence of a direct effect of GDF5 on muscle[36].

The locus on Chr 18 was near to the DYM gene. Loss of function mutations in this gene are associated with Dyggve-Melchior-Clausen syndrome and Smith-McCort dysplasia respectively. Mice lacking this gene present with chondrodysplasia resulting from impaired endochondral bone formation and abnormalities of the growth plate that begin to manifest shortly birth[37]. In homozygous mutant mice, and in patients with loss of function mutations in this gene, both the axial and the appendicular skeleton are affected. As is noted in Supplementary Data 2, this gene is also associated with height in the GWAS catalogue. Interestingly, this gene is highly expressed in skeletal muscle in humans[38], however its function in muscle is not completely understood.

SLC39A8 encodes for the metal ion transporter ZIP8 which has been shown to be upregulated in chondrocytes present in osteoarthritic cartilage[39]. The lead SNP rs13107325 is a missense variant within SLC39A8 which has been previously associated with osteoarthritis[40].

On chromosome 12 genetic variants known to affect expression of MGP (Matrix Gla Protein) in the tibial nerve (among other tissues) are associated with osteoarthritis of the hand but not of the hip or knee[41]. MGP is an inhibitor of arterial and soft tissue calcifications, with links to atherosclerosis[42]. Consistent with this, older women with severe abdominal aortic calcification have greater decline in grip strength over 5 years[43]. More recently a study suggested MGP may also regulate muscle development and atrophy[44].

We also identified variants known to affect Transforming Growth Factor Alpha (TGFA) expression (increased in the testis, decreased in the brain), which is implicated in cell proliferation, differentiation and development. This locus has previously been identified for overall strength[10], and suggestive evidence from a study of 1,323 participants linked rs2862851, a variant in linkage disequilibrium with the lead SNP rs958685 ($R^2 = 0.90$, D' = 1.0), with increased risk of osteoarthritis in the knee (OR = 1.4, $p = 3.1 \times 10^{-4}$)[45].

The low grip strength locus on chromosome 15 is near ALDH1A2, which has a key role in the pathogenesis of osteoarthritis[46]. Low grip strength-associated variants at this locus have previously been identified for severe osteoarthritis of the hand, and may explain why this locus is associated with low grip strength measured by hand dynamometer[46]. Knocking out this gene in mice is perinatally lethal, however at embryonic day 18.5 mice lacking this gene do present with numerous cartilage gene defects[47]. We also identified variants that affect expression of CHRDL2 (Chordin Like 2) in the thyroid, known to interact with mouse Gdf5, which is upregulated in human osteoarthritic joint cartilage cell line[48]. In addition, down-regulation of CHRDL2 expression has been linked to the progression of severe osteoarthritis in the knee joint[49], suggesting a role for CHRDL2 in cartilage repair.

Two of the identified loci (RBBP6 'RB Binding Protein 6, Ubiquitin Ligase' and ZBTB38 'Zinc Finger And BTB Domain Containing 38') form an axis involved in DNA replication and chromosomal stability[50]. RBBP6 ubiquitinates the transcriptional repressor ZBTB38, destabilizing it and reducing its action on the replication factor MCM10. In mice, Zbtb38 is highly expressed in skeletal muscle, loss of this methyl-CpG-binding protein (which is also known as Cibz), promotes myogenic differentiation. Conversely, expression of Zbtb38 is decreased in satellite cells during muscle regeneration[51], suggesting that like other members of this gene family, this gene is involved in cellular differentiation. Variants associated with low grip strength in our analysis are known to decrease ZBTB38 expression in whole blood and skeletal muscle (among others) and increase expression in the skin.

Mitochondrial dysfunction is a hallmark of aging, yet we found no variants associated with low strength at genome-wide significance levels in a sub-analysis of UK Biobank only. Variants in MT-CYB were nominally significant (p = 0.0003): MT-CYB (mitochondrial cytochrome b) is part of the mitochondrial respiratory chain, and essential for Complex III formation. Monogenic diseases associated with MT-CYB include exercise intolerance and "Additional features include lactic acidosis, muscle weakness and/or myoglobinuria" (https://www.uniprot.org/uniprot/P00156). Recent work by Cohen et al. have highlighted the importance of mitochondrial peptides such as humanin in many age-related diseases[52], however we found no variants in these genes associated with low grip strength passing multiple testing correction.

In comparison to a previous study that analysed grip strength as a continuous measure in the UK Biobank cohort[9] that included

all aged (40–70 years), we found that only three of the 64 identified variants were significantly ($p < 5 \times 10^{-8}$) associated with EWGSOP low grip strength (Supplementary Data 2). The top association was rs13107325 (*SLC39A8* linear grip $p = 4.4 \times 10^{-23}$, low grip strength meta-analysis $p = 7.4 \times 10^{-11}$), then rs2430740 (*C12orf60* linear grip $p = 6 \times 10^{-12}$, low grip strength meta-analysis $p = 2.6 \times 10^{-9}$) and finally rs11236203 (*POLD3*, linear grip $p = 8.4 \times 10^{-10}$, low grip strength meta-analysis $p = 2.8 \times 10^{-8}$). Two less-significant associations ($<1 \times 10^{-6}$) were seen with rs3821269 (*TGFA* linear grip $p = 3.5E \times 10^{-15}$, low grip strength meta-analysis $p = 8.0 \times 10^{-8}$) and rs1556659 (*ENSG00000232985* linear grip $p = 1.1 \times 10^{-11}$, low grip strength meta-analysis $p = 3.5 \times 10^{-7}$). A previous GWAS by the CHARGE consortium identified two loci associated with maximum hand grip strength recorded in 27,581 Europeans aged 65 or older[8]. We found that neither locus was associated with low grip strength in our analysis (rs752045 EWGSOP $p = 0.67$, rs3121278 EWGSOP $p = 0.15$).

We observed minimal overlap between loci associated with low grip strength and general anthropometric traits such as height and continuous measures of strength. The SNP-based heritability estimate for EWGSOP low grip strength in older adults was 4.4% (SE 0.3%). This was somewhat lower than the 13% (SE 0.4%) SNP-based heritability for continuous grip strength reported in UK Biobank participants aged 40–70[9]. This may be partly explained by our study using a binary cut-off for low grip compared to the quantitative analysis of grip strength. These results emphasize that the genetics of muscle weakness and overall strength are distinct. The SNP-based heritability estimates observed here are lower than those from studies of twins, for example a study of 1,757 male twin pairs aged 45–96 found the heritability of continuous strength to be 48–55%[7], however other studies have shown that heritability declines significantly as age advances as environmental factors explain more of the variance[3], though still up to 22% has been reported for muscle strength. In contrast to the twin studies our estimates of heritability are restricted to common SNPs with MAF ≥ 1%, and thus represent a lower bound of the overall genetic variance of low strength in older people.

Despite seeing little overlap at the individual locus level between low grip strength at older ages and other diseases we did observe significant genetic correlations, especially with osteoarthritis (30% overlap). However in Mendelian Randomization analysis we did not observe a causal relationship between osteoarthritis and low grip strength: taken together, this suggests that osteoarthritis shares causal risk factors and biological pathways with low grip strength at older ages, such as obesity, but may not cause it. In addition, osteoarthritis has diverse loci associated with different joints, for instance osteoarthritis in the hand appears to have a distinct genetic signal: the low grip strength locus associated with *MGP* expression has been previously linked to osteoarthritis in the fingers and hand, but not the hip or knee[41]. Although our results were robust to adjustment for osteoarthritis (including rhizarthrosis—arthritis of the thumb) this may suggest that arthritis in the hand needs to be accounted for in measures of muscle weakness, as hand grip strength may not always reflect muscle strength elsewhere, e.g., lower-extremity strength.

Our Mendelian Randomization analyses highlighted specific traits and diseases which may share causal pathways with weakness at older ages. This included growth and development traits such as birth weight, waist:hip ratio, and pubertal timing (age at menarche) in women (highly genetically correlated – 75% – with age at voice breaking in men[53]), where greater values were protective. Although puberty timing is highly polygenic, it is strongly genetically correlated with BMI ($-35\%$)[53], with complex

interactions: i.e., being thinner in childhood is associated with delayed menarche[54], but later menarche results in taller adult height[55]. These results are consistent with the observation that growth and development traits are associated with strength trajectories in later life[56]. We chose to not adjust for body size in our primary analysis in case interesting or novel effects were masked, but in follow-up analysis determined that all except two of the loci identified were predominantly independent of participant height and weight.

Raised red blood cell parameters - especially plateletcrit, the proportion of blood occupied by platelets - appear to be protective in males but associations were attenuated or nonsignificant in females. A number of studies have recently reported a link between raised platelet counts, inflammation, and sarcopenia cross-sectionally[57]. However our results suggest that plateletcrit across the lifecourse (rather than after sarcopenia onset) may be different. Lastly, only four of the conditions we investigated (which included coronary artery disease, and some common cancers) appear to causally increase risk of weakness in older people: depression, asthma/allergic diseases, Rheumatoid arthritis, and type-2 diabetes. These are diverse conditions and further underlines the multifactorial causes of weakness in older people.

There are a number of limitations to our analyses. The data included are predominantly from subjects at the younger end of the age 60 plus demographic, and is not enriched for frail individuals. The sample size for sex-specific analyses is limited, especially for men, likely contributing to the fewer significant associations observed. The results from sex-stratified analyses need to be interpreted with caution, as a recent pre-print on bioRxiv has shown that some variants are spuriously associated with sex in cohorts such as the UK Biobank, likely due to their effect on differential participation[58]. Our analysis was limited to relatively common variants (prevalence >1%) in subjects with European ancestries only. The analyses of the FNIH strength cutpoints also have limited power, given the low prevalence of the phenotype, although studying the extremes of a continuous trait can provide increased power if stronger associations are uncovered. We did not include additional analysis of the revised EWGSOP2 low grip cut-points[6] as these are almost identical to the FNIH criteria, which many cohorts had already analyzed; future analyses should include this. Analysis of rare and structural variants, and analyses in other ancestral groups will give a more complete picture of the genetic landscape for low grip strength or dynapenia. Some Mendelian Randomization analyses have limited power due to the lack of strong instruments, and therefore null results for these analyses should be interpreted with caution.

To conclude, genetic variation in 15 loci are related to muscle weakness in people aged 60 plus, of European descent, with limited overlap with loci associated with the full range of muscle strength in 40–70 year olds. The loci implicated may be involved in hallmark pathways of ageing including cell cycle control and inflammation, along with loci implicated in arthritis and pathways involved in the development and maintenance of the musculoskeletal system.

## Methods

**GWAS of low grip strength in older people.** We conducted a GWAS meta-analysis of low grip strength in participants aged 60 years or older of European ancestry from 22 studies yielding a combined sample of 254,894 individuals. Individual studies used different genotyping platforms and imputation was predominantly performed using the Haplotype reference consortium (HRC) v1.1 panel. See Supplementary Information for details on individual study methods.

Two definitions of low muscle hand grip strength were utilized at the time of analysis. The primary analysis was of the 2010 EWGSOP criteria for sarcopenic grip strength (Grip strength <30 kg Male; <20 kg Female). In secondary analysis we considered a more data-driven definition with more strict thresholds by the FNIH sarcopenia project 2014 (Grip strength <26 kg Male; <16 kg Female) for

comparison. Given the known differences in strength between males and females (on average) we also performed sex stratified analyses.

GWAS was performed by each cohort individually (see Supplementary Methods) using regression models, adjusted for age, sex (except in sex-specific models), and population substructure, accounting for relatedness and technical covariates as required by the individual study. No adjustment for anthropometric measures was made in the primary analysis, but the effects were explored in sensitivity analyses (see below). Fixed-effects inverse variance weighted meta-analysis was performed using METAL[59] using the GWAS summary statistics generated by each cohort, with genomic control for population structure (see Supplementary Methods for details). The following quality control filters were applied: minor allele frequency (MAF) > 0.01, imputation info score of > 0.4, and the variant present in at least two studies (UK Biobank – the largest included cohort—plus at least one other). The final analysis therefore included 9,678,524 genetic variants. Associations that achieved a $p < 5 \times 10^{-8}$ were considered statistically significant, with those reaching the more stringent threshold of $p < 5 \times 10^{-9}$ highlighted.

Distinct loci were initially defined as two significant variants separated by >500 kb. To identify independent signals at each locus we used FUMA (Functional Mapping and Annotation of Genome-Wide Association Studies)[60], which uses Linkage Disequilibrium (LD) information to determine independence ($r^2$ threshold = 0.1 for independent significant SNP). We used Linkage Disequilibrium Score Regression (LDSC, v1.0.0) to estimate the level of bias (i.e., from population stratification and cryptic relatedness) in the GWAS, and the SNP-based heritability of low grip strength[13].

**Locus overlap with diseases and anthropometric traits**. The GWAS catalogue of published loci-trait associations[11] was searched to identify whether low grip strength-associated loci determined from our meta-analysis are known to influence other traits or diseases. In addition, we performed sensitivity analyses in the UK Biobank sample to determine whether associations between variants and low grip strength identified in the meta-analysis were robust to adjustment for the following traits or diseases: height, weight, body mass index, osteoarthritis, rheumatoid arthritis, and osteoporosis (prevalent diseases were from the self-reported data at baseline or in the Hospital Episode Statistics). Analyses were performed in STATA (v 15) using logistic regression models adjusted for age, sex, principal components 1 to 10, assessment centre, and genotyping array (the UK Biobank using two different Affymetrix microarrays that shared >95% of sites – see Supplementary Methods).

**Gene ontology pathways, tissue enrichment, and eQTL analyses**. We utilized FUMA to perform functional interpretation of the GWAS results[60]. In particular, FUMA performs gene-set analysis (using Multi-marker Analysis of GenoMic Annotation (MAGMA)[61]) to identify pathways enriched amongst the significant genes (weighted by the SNP-associations in proximity to them), in addition to searching eQTL databases to identify SNPs that significantly alter the expression of genes in various tissues.

We used MetaXcan to determine whether gene-level transcriptomic associations in GTEx v7 skeletal muscle data were enriched in the GWAS summary statistics for low grip strength[17]. The analysis included 7512 genes with measured expression in the dataset; we applied Benjamini-Hochberg multiple-testing correction, with adjusted $p$ values < 0.05 deemed to be significant.

LD Score Regression applied to specifically expressed genes (LDSC-SEG) allows the identification of enriched tissue activity associated with GWAS results[18]. We applied LDSC-SEG (v1.0.0) to the GWAS summary statistics using the datasets 'Multi_tissue_gene_expr' and 'Multi_tissue_chromatin' provided by the authors. We applied Benjamini-Hochberg multiple-testing correction for the 703 tests (n gene expression = 205, n chromatin = 498), with adjusted $p$ values < 0.05 deemed to be significant.

**Genetic correlations and Mendelian randomization**. We investigated the genetic correlation between the low grip strength trait and 10 diseases - chosen because they are common, chronic diseases of aging - using LDSC (v1.0.0)[13] and published GWAS summary statistics for the following diseases: Alzheimer's disease[62], breast cancer[63], chronic kidney disease[64], coronary artery disease[65], osteoporotic fracture risk[66], osteoarthritis[67], prostate cancer[68], rheumatoid arthritis[69], stroke[70], and type-2 diabetes[71]. We also calculated genetic correlations with the following anthropometric traits: height[72], body mass index (BMI)[72], waist:hip ratio (WHR)[73], whole-body lean mass[12], and appendicular lean mass[12].

We also undertook a phenotype-wide Mendelian randomization (MR) association study to examine the causal effect of 83 traits on low hand grip strength. We used the "TwoSampleMR" (v0.4.23) package in R[19] to perform the analysis of genetic instruments from the 83 traits, which including those traits with clear biological rationale (for example, adiposity) and others that are more exploratory for hypothesis generation (for example, puberty timing). Follow up sensitivity analysis of the identified traits was by the MR-Egger and using weighted median estimation methods provided in the package.

**Reporting summary**. Further information on research design is available in the Nature Research Reporting Summary linked to this article.

## Data availability
The GWAS summary statistics and supporting information on low grip strength in older people are available on the Musculoskeletal Knowledge Portal (www.mskkp.org) and the GWAS catalogue (www.ebi.ac.uk/gwas accession numbers GCST90007526, GCST90007527, GCST90007528, GCST90007529, GCST90007530 and GCST90007531). The International Mouse Phenotyping Consortium database is located (https://www.mousephenotype.org/). eQTL data is available from (https://gtexportal.org/). Catalogue of GWAS associations is available (https://www.ebi.ac.uk/gwas/). All relevant additional data is available on request from the authors. Information on the 22 individual cohorts is included in the Supplementary Information file.

## Code availability
See our associated GitHub repository for example scripts used for the main analyses (https://github.com/pasted/gw_meta_analysis_low_muscle_strength). All relevant additional code is available on request from the authors.

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

## Acknowledgements

This study was part funded by an award to DM by the UK Medical Research Council (MR/M023095/1). A full list of acknowledgements and grant support can be found in Supplementary Note 1 (in the Supplementary Information). We would like to acknowledge the use of the University of Exeter High-Performance Computing (HPC) facility in carrying out this work.

## Author contributions

G.J., K.T., A.J.S., N.S., C-LK., and J.L.A. contributed equally to this work. G.A.K., L.F., D.K., F.R., D.K.K., and L.C.P. jointly supervised this work. D.E.A., B.G.W., E.B., M.L.G., M.G., I.D., N.v.dV, L.C.P.G.M.d.G, B.M.P., C.L., N.J.W., S.B., N.M.v.S, M.H, J.M.Z., D.M., M.K., D.A.B., A.S.B., P.L.D.J., A.G.U., A.T., L.R.M., F.G.G., J.C., P.H., L.B., C.O., J.M.M, L.F., D.K., F.R., and D.P.K. designed and managed individual studies. B.G.W., N.M.v.S, M.H, D.A.B., A.S.B., P.L.D.J., L.R.M., F.G.G., J.C., and P.H. collected data. G.J., K.T., A.J.S., N.S, J.R.L, J.M.M, G.A.K, L.F., D.M., D.K., F.R., D.P.K. and L.C.P. reviewed the analysis plan. G.J., K.T., A.J.S., N.S, C.K., T.V.D., S.H., M.L.B., J.L., C.S., K. L. L., T.T, M.K.W., R.C., M.N., S.G., J.Y., M.C., S.W., K.S., A.T., and L.C.P. analyzed the data. G.J. and L.C.P. performed the meta-analysis. G.J., C.A-B and L.C.P. performed the pathway and other analyses. J.M.M, G.A.K., L.F., D.M., D.K., F.R., D.P.K., and L.C.P. supervised the overall study design. G.J., K.T., A.J.S., N.S, C.K., J.L.A., J.M.M, G.A.K, L.F., D.M., D.K., F.R., D.P.K. and L.C.P. wrote the manuscript. G.J., K.T., A.J.S., N.S, C.K., J.L.A., J.R.L, T.V.D., S.H., M.L.B., J.L., C.S., K. L. L., T.T, M.K.W., R.C., M.N., S.G., J.Y., M.C., S.W., K.S., E.K., C.A-B., D.E.A., B.G.W., E.B., M.L.G., M.G., D.S., I.D., N.v.dV, L.C.P.G.M.d.G, B.M.P., M.C.O., A.E.F., C.L., N.J.W., S.B., N.M.v.S, M.H, Q.T., J.M.Z., D.M., M.K., D.A.B., A.S.B., P.L.D.J., A.G.U., U.V., T.K., A.T., L.R.M., F.G.G., J.C., P.H., L.B., C.O., J.M.M, G.A.K, L.F., D.M., D.K., F.R., D.P.K. and L.C.P. reviewed the manuscript.

## Competing interests

D.P.K. is a consultant for Solarea Bio, received institutional grants from Amgen and Radius Health, and received royalties for publication Wolters Kluwer. M.C.O. serves as a consultant for Cricket Health, a kidney care company. B.M.P. serves on the Steering Committee of the Yale Open Data Access Project funded by Johnson & Johnson. All other authors declare no competing interests.

## Additional information

[1]Epidemiology and Public Health Group, Institute of Biomedical and Clinical Science, University of Exeter Medical School, Exeter, UK. [2]Department of Internal Medicine, Erasmus Medical Center, Rotterdam, The Netherlands. [3]Department of Epidemiology Medicine, Erasmus Medical Center, Rotterdam, The Netherlands. [4]University of Pittsburgh, Department of Epidemiology, Pittsburgh, PA, USA. [5]Department of Epidemiology and Biostatistics, Amsterdam UMC- Vrije Universiteit, Amsterdam Public Health Research Institute, Amsterdam, The Netherlands. [6]Biostatistics Center, Connecticut Convergence Institute for Translation in Regenerative Engineering, UConn Health, Farmington, CT, USA. [7]School of Medical and Health Sciences, Edith Cowan University, Joondalup, WA, Australia. [8]School fo Public Health University of Sydney, Sydney, NSW, Australia. [9]Medical School, University of Western Australia, Crawley, WA, Australia. [10]McKusick-Nathans Institute, Department of Genetic Medicine, Johns Hopkins University School of Medicine, Baltimore, MD, USA. [11]Lübeck Interdisciplinary Plattform for Genome Analytics, Institutes of Neurogenetics and Cardiogenetics, University of Lübeck, Lübeck, Germany. [12]Cardiovascular Health Research Unit, Department of Medicine, and Department of Biostatistics, University of Washington, Seattle, WA, USA. [13]MRC Epidemiology Unit, Institute of Metabolic Science, University of Cambridge School of Clinical Medicine, Cambridge CB2 0QQ, UK. [14]Biostatistics Department, Boston University School of Public Health, Boston, MA, USA. [15]Longitudinal Study Section, Translational Gerontology branch, National Institute on Aging, Baltimore, MD, USA. [16]Department of Genetics, Washington University School of Medicine, St. Louis, MO, USA. [17]Centre for Bone and Arthritis Research, Department of Internal Medicine and Clinical Nutrition, Institute of Medicine, Sahlgrenska Academy, University of Gothenburg, Gothenburg, Sweden. [18]Bioinformatics Core Facility, Sahlgrenska Academy, University of Gothenburg, Gothenburg, Sweden. [19]Department of Psychiatry and Psychotherapy, University Medicine Greifswald, Greifswald, Germany. [20]Institute for Community Medicine, University Medicine Greifswald, Greifswald, Germany. [21]Rush Alzheimer's Disease Center & Department of Neurological Sciences, Rush University Medical Center, Chicago, IL, USA. [22]Department of Medicine and Public Health, Rey Juan Carlos University, Madrid, Spain. [23]CIBER of Frailty and Healthy Aging (CIBERFES), Madrid, Spain. [24]Center for Demography of Health and Aging, University of Wisconsin-Madison, Madison, WI, USA. [25]School of Physiology, Pharmacology and Neuroscience, University of Bristol, Bristol, UK. [26]Department of Orthopedics, University of Colorado, Aurora, CO, USA. [27]Department of Medicine/Geriatrics, University of Mississippi School of Medicine, Jackson, MS, USA. [28]Human Genetics Center, Department of Epidemiology, Human Genetics, and Environmental Sciences, School of Public Health, The University of Texas Health Science Center at Houston, Houston, TX, USA. [29]Human Genome Sequencing Center, Baylor College of Medicine, Houston, TX, USA. [30]Department of Epidemiology, University of North Carolina, Chapel Hill, NC 27516, USA. [31]Charité - Universitätsmedizin Berlin, corporate member of Freie Universität Berlin, Humboldt-Universität zu Berlin, Berlin, Germany. [32]Berlin Institute of Health, Department of Endocrinology and Metabolism, Berlin, Germany. [33]Charité - Universitätsmedizin Berlin, BCRT - Berlin Institute of Health Center for Regenerative Therapies, Berlin, Germany. [34]Department of Internal Medicine, Section of Geriatric Medicine, Academic Medical Center, University of Amsterdam, Amsterdam, The Netherlands. [35]Wageningen University, Division of Human Nutrition, PO-box 17, 6700 AA Wageningen, The Netherlands. [36]Cardiovascular Health Research Unit, Departments of Medicine, Epidemiology, and Health services, University of Washington, Seattle, WA, USA. [37]Kaiser Permanente Washington Health Research Institute, Seattle, WA, USA. [38]Department of Epidemiology and Population Health, Stanford University, Stanford, CA, USA. [39]Department of Epidemiology and Institute of Public Genetics, University of Washington, Seattle, WA, USA. [40]Geriatric Unit, Azienda Sanitaria Firenze (ASF), Florence, Italy. [41]Epidemiology and Biostatistics, Department of Public Health, Faculty of Health Science, University of Southern Denmark, Odense, Denmark. [42]Geriatric Medicine, Institute of Medicine, Sahlgrenska Academy, University of Gothenburg, Gothenburg, Sweden. [43]Clinical and Molecular Osteoporosis Research Unit, Department of Orthopedics and Clinical Sciences, Lund University, Skåne University Hospital, Malmö, Sweden. [44]Center for Translational and Systems Neuroimmunology, Department of Neurology, Columbia University Medical Center, New York, NY, USA. [45]Program in Medical and Population Genetics, Broad Institute, Cambridge, MA, USA. [46]Interfaculty Institute for Genetics and Functional Genomics, University Medicine Greifswald, Greifswald, Germany. [47]Department of Restorative Dentistry,

Periodontology, Endodontology, and Preventive and Pediatric Dentistry, University Medicine Greifswald, Greifswald, Germany. [48]Department of Geriatrics, Getafe University Hospital, Getafe, Spain. [49]Department of Geriatrics, Hospital Virgen del Valle, Complejo Hospitalario de Toledo, Toledo, Spain. [50]Professor of Public Policy, Georgetown University, Washington, DC, USA. [51]Sahlgrenska University Hospital, Department of Drug Treatment, Gothenburg, Sweden. [52]Section of General Internal Medicine, Boston University School of Medicine, Boston, MA, USA. [53]Center on Aging, University of Connecticut Health, 263 Farmington Avenue, Farmington, CT 06030, USA. [54]National Institute on Aging, Baltimore, MD, USA. [55]Marcus Institute for Aging Research, Hebrew SeniorLife, Boston, MA, USA. [56]Azrieli Faculty of Medicine, Bar Ilan University, Safed, Israel. [57]Marcus Institute for Aging Research, Hebrew SeniorLife and Department of Medicine, Beth Israel Deaconess Medical Center and Harvard Medical School, Broad Institute of MIT & Harvard, Boston, MA, USA. [58]These authors contributed equally: Garan Jones, Katerina Trajanoska, Adam J. Santanasto, Najada Stringa, Chia-Ling Kuo, Janice L. Atkins. [59]These authors jointly supervised this work: George A. Kuchel, Luigi Ferrucci, David Karasik, Fernando Rivadeneira, Douglas P. Kiel, Luke C. Pilling. ✉email: L.Pilling@exeter.ac.uk

