## [Peer Review File · Nature Communications]

REVIEWER COMMENTS

Reviewer #1 (Remarks to the Author):

Nature Communications peer review report

Genome-wide meta-analysis of muscle weakness identifies 15 susceptibility loci in older men and women

Jones et al.

This article describes a GWAS meta-analysis of muscle weakness using a binary case:control analysis in contrast to previous work that has defined muscle strength as a continuous variable. The authors discovered 15 loci, 12 of which have not previously been identified in the continuous variable analyses.

I enjoyed reading the paper. The work is novel, and will be of interest to the scientific community. The paper overall is very well-written. I have suggested changes that would improve clarity of the manuscript, and one potential re-analysis. I would suggest acceptance after satisfactory addressing of these comments.

Comments

1. Throughout, an asterisk is used instead of a multiplication symbol. Please replace.
2. In the introduction, you state muscle strength is heritable. Please expand. What is the heritability from twin studies?
3. Results: "Twelve of the fifteen lead SNPs from the GWAS have not previously been identified in studies of continuous grip strength in all ages 10(Table 2)." Add in: "Also, only X of XX previously reported continuous hand grip strength loci were associated with low grip strength" (I think this is 3 of 64 from Supp table 6).
4. Results: "other independent variants ($R^2 < 0.6$) at the same locus (e.g. rs3204689) have been found to be associated with osteoarthritis (Supplementary Table 2)" – the ALDH1A2 variants are specifically Hand OA – this could be why you picked these up if the low grip strength (measured on a dynamometer) is caused by OA in the hand. Please discuss this as a possibility, or account for it by doing an adjusted analysis (I suspect this data may not be available to you). See also point 14 below.
5. Results: "The single nucleotide polymorphism (SNP) based heritability (h^2) of low grip strength was 0.044 (SE 0.0027), i.e. 4.4%, by LD Score Regression." How does this compare to heritability from twin studies?
6. Results: "for example rs201754 (chr13:51078446, Female only) and rs3118903 (chr13:51099577, both sexes) with $D' = 0.97$ and $r^2 = 0.91$." Remove this example – it adds nothing, and readers understand LD.
7. Results: "rs35225200, was in linkage disequilibrium with a lead SNP from the combined analysis (rs13107325 $R^2 = 0.89$, $D' = 1.0$). While the" Remove this.
8. Results: "The reduced number of cases in males could contribute to the fewer associations observed, and the borderline significance (p -values close to 5×10^{-8})." Remove this.
9. Results: "The results from sex-stratified analyses need to be interpreted with caution, as a recent pre-print on bioRxiv has shown that variants are associated with sex in cohorts such as the UK Biobank, likely due to their effect on participation 13." This is not a result, it is a discussion. Move.
10. Results: "Analysis of 116 mitochondrial genetic variants ($MAF > 0.01$) available in the UK Biobank directly genotyped microarray data identified two associated with EWGSOP-defined low hand grip strength ($p < 0.00043$, i.e. Bonferroni-adjustment for mitochondrial variants).

rs41518645 is a missense variant (p.Asp171Asn) in MT-CYB, identified in Plink logistic regression analysis ($p=0.0003$). rs201950015 is intronic, located between genes CO1 and ATP6/8 ($p=0.00042$). No variants achieved "genome-wide" significant ($p<5*10^{-8}$). See Supplementary Table 5." I suggest you remove the mitochondrial analysis from the paper. There is no replication or meta-analysis, and it adds nothing to the story. The poor methodological quality detracts from the rest of the paper.

11. Results section: Subset of overall grip strength genetic variants are associated with low grip strength – this is not really results, more discussion of overlap between studies. Suggest you report in point 3 above, and move this to discussion.

12. Results: "although genes expressed in the tibial nerve were borderline enriched (Benjamini-Hochberg-adjusted p-value 0.1)." – remove this. Borderline is not significant. See <https://royalsocietypublishing.org/doi/10.1098/rsos.140216> for an elegant description of why.

13. Results: "To explore the effect of genetic predisposition to low grip strength at older ages we created an unweighted genetic risk score (GRS) in the UK Biobank European sample" Why did you use an unweighted score, rather than the more usual weighted GRS?

14. Results: "co-morbidities we performed regression analyses in the UK Biobank cohort with adjustment for the following covariates: height, weight, skeletal muscle mass (determined using bioimpedance analysis), osteoarthritis, Rheumatoid arthritis, osteoporosis, Dupuytren's contracture (one or more fingers permanently bent), and rhizarthrosis (arthritis of the thumb)." How did you determine if a person was affected by OA, RA, OP, DD, and thumb arthritis? Was this just on ICD10 codes, or did you use OPCS codes too (for example for hip or knee replacement, DD surgery etc.) and self-report? If you only used ICD-10 you will have missed a significant proportion of cases, and will need to re-run this analysis. Please also provide a supplementary table of the exact codes you have used for this analysis.

15. Discussion: "In this study of 256,523 Europeans aged 60 years and over we identified 15 genetic loci - 3 novel - associated with the EWGSOP definition of low grip strength (dynapenia)," I thought 12 were novel?

16. Discussion: "Only three of these were previously implicated in previous analyses of continuous strength measures, suggesting that the genetic causes of clinically meaningful weakness at older ages are partly distinct." Again, add in the converse to emphasise the lack of meaningful overlap (see point 3).

17. Discussion: "of age-related diseases HLA-DQA1 was confirmed" confirmed is not the right word here, perhaps associated or identified?

18. Discussion: "Overall five of the fifteen genomic risk loci for EWGSOP low grip strength have been previously associated, or in linkage disequilibrium with variants associated with osteoarthritis" Did you miss out the ALDH1A2 locus here?

19. Discussion: "adiposity, a known risk factor for arthritis 28." Please use osteoarthritis, not generic arthritis, which may include Rheumatoid.

20. Discussion: "The lack of this gene in mice is perinatally lethal, however at emb" Change to knockout.

21. Discussion: "Mitochondrial dysfunction is a hallmark of aging, yet we found no variants..." Suggest removal of this paragraph.

22. Discussion: "Although puberty timing is highly polygenic, it is strongly genetically correlated with BMI" Later menarche causes taller height (which accounts for at least some of the association with lower BMI). This is a complex inter-related area, and this should be brought out in the discussion.

23. Discussion: "we future analyses should include this." Not correct English.

24. Table 1. Please add whether the reported SNP is genotyped or imputed. If imputed, please add the info score. I am concerned that the info score cut off was too lax – 0.9 is a better threshold, but it is OK if the associated imputed SNPs have an info score of >0.9 .

25. Table 1. Please add the effect allele frequency.

26. Table 1. The headers and rsID numbers are in bold. As are the p-values below $5*10^{-9}$. I don't find this useful, they are not special. I would prefer the entire row of the novel variants to be highlighted in bold.

27. Table 2. Suggest this goes in Supplementary.

Reviewer #2 (Remarks to the Author):

This is a very well written manuscript of a GWAS meta-analysis of muscle weakness that employs a large number of cases and controls aged 60 years and over. Several novel loci have been identified, with subsequent comparisons with other age-related conditions and with anthropometric traits highlighting examples of genetic overlap. There are also the standard pathway analyses, searches for eQTL effects, MR analysis, etc. One of the striking genetic correlations was with OA. This is intriguing and the authors are commended for noting that a proportion of this may be the impact that painful hand OA will have on the ability to adequately undertake a grip strength test. Overall, I think there are some novel and interesting insights reported, the investigators have undertaken all necessary analyses, and they have drawn reasonable and logical conclusions. I do not have any changes to recommend.

Reviewer #1 (Remarks to the Author):

This article describes a GWAS meta-analysis of muscle weakness using a binary case:control analysis in contrast to previous work that has defined muscle strength as a continuous variable. The authors discovered 15 loci, 12 of which have not previously been identified in the continuous variable analyses.

I enjoyed reading the paper. The work is novel, and will be of interest to the scientific community. The paper overall is very well-written. I have suggested changes that would improve clarity of the manuscript, and one potential re-analysis. I would suggest acceptance after satisfactory addressing of these comments.

Comments

1. Throughout, an asterisk is used instead of a multiplication symbol. Please replace.

We have changed these throughout the manuscript.

2. In the introduction, you state muscle strength is heritable. Please expand. What is the heritability from twin studies?

We have added a reference to the following study of 1,757 male twin pairs aged 45-96 by Frederiksen et al. "Hand grip strength: A phenotype suitable for identifying genetic variants affecting mid- and late-life physical functioning" – text now reads:

Importantly, muscle strength is heritable (48-55% in 1,757 male twin pairs aged 45-96)⁷, and can thus be used in genetic investigations.

3. Results: "Twelve of the fifteen lead SNPs from the GWAS have not previously been identified in studies of continuous grip strength in all ages 10(Table 2)." Add in: "Also, only X of XX previously reported continuous hand grip strength loci were associated with low grip strength" (I think this is 3 of 64 from Supp table 6).

We have extended the sentence to now read:

Twelve of the fifteen lead SNPs from the GWAS have not previously been identified in studies of grip strength on a continuous scale across all ages (Supplementary Tables 3 & 4) and only 3 of the 64 loci associated with overall muscle strength 10 are significant in our analysis of low strength (Supplementary Table 5).

4. Results: "other independent variants ($R^2 < 0.6$) at the same locus (e.g. rs3204689) have been found to be associated with osteoarthritis (Supplementary Table 2)" – the ALDH1A2 variants are specifically Hand OA – this could be why you picked these up if the low grip strength (measured on a dynamometer) is caused by OA in the hand. Please discuss this as a possibility, or account for it by doing an adjusted analysis (I suspect this data may not be available to you). See also point 14 below.

Unfortunately, an adjusted analysis was not possible. However, we agree with the reviewer that this explanation is plausible. We have extended our discussion of ALDH1A2 in the discussion section to explicitly reference this point:

The low grip strength locus on chromosome 15 is near ALDH1A2, which has a key role in the pathogenesis of osteoarthritis⁴⁵. Low grip strength-associated variants at this locus have previously been identified for severe osteoarthritis of the hand, and may explain why this locus is associated with low grip strength measured by hand dynamometer⁴⁵.

5. Results: “The single nucleotide polymorphism (SNP) based heritability (h^2) of low grip strength was 0.044 (SE 0.0027), i.e. 4.4%, by LD Score Regression.” How does this compare to heritability from twin studies?

We have added the following to the Discussion section of the heritability estimates:

*We observed minimal overlap between loci associated with low grip strength and general anthropometric traits such as height and continuous measures of strength. The SNP-based heritability estimate for EWGSOP low grip strength in older adults was 4.4% (SE 0.3%). This was somewhat lower than the 13% (SE 0.4%) SNP-based heritability for continuous grip strength reported in UK Biobank participants aged 40 to 70⁹. This may be partly explained by our study using a binary cut-off for low grip compared to the quantitative analysis of grip strength. These results emphasize that the genetics of muscle weakness and overall strength are distinct. **The SNP-based heritability estimates observed here are lower than those from studies of twins, for example a study of 1,757 male twin pairs aged 45-96 found the heritability of continuous strength to be 48-55%⁷, however other studies have shown that heritability declines significantly as age advances as environmental factors explain more of the variance³, though still up to 22% has been reported for muscle strength. In contrast to the twin studies our estimates of heritability are restricted to common SNPs with MAF $\geq 1\%$, and thus represent a lower bound of the overall genetic variance of low strength in older people.***

6. Results: “for example rs201754 (chr13:51078446, Female only) and rs3118903 (chr13: 51099577, both sexes) with $D' = 0.97$ and $r^2 = 0.91$.” Remove this example – it adds nothing, and readers understand LD.

We have removed the example of correlated variants.

7. Results: “rs35225200, was in linkage disequilibrium with a lead SNP from the combined analysis (rs13107325 $R^2 = 0.89$, $D' = 1.0$). While the” Remove this.

We have removed the reference to LD with a lead SNP from the main analysis, and reworded that section as follows:

The analysis of males only (total $N = 118,371$ with 13,327 cases, 11.3%) identified three genomic loci associated with the EWGSOP low grip strength definition. Two of these variants appeared to be distinct signals from the overall analysis and were not associated with low grip in females (see Supplementary Table 7 for details): rs774787160 mapped to gene DSCAM (males $p = 1 \times 10^{-8}$; females $p = 0.9$) and rs145933237 mapped to mir-466, which was only nominally associated in females (males $p = 2 \times 10^{-8}$; females $p = 0.01$).

8. Results: “The reduced number of cases in males could contribute to the fewer associations observed, and the borderline significance (p -values close to 5×10^{-8} .” Remove this.

We have removed the sentence from the results.

9. Results: “The results from sex-stratified analyses need to be interpreted with caution, as a recent pre-print on bioRxiv has shown that variants are associated with sex in cohorts such as the UK Biobank, likely due to their effect on participation 13.” This is not a result, it is a discussion. Move.

We have moved this section to the limitation paragraph in the Discussion. It now reads:

The sample size for sex-specific analyses is limited, especially for men likely contributing to the fewer significant associations observed. The results from sex-stratified analyses need to be interpreted with caution, as a recent pre-print on bioRxiv has shown that some variants are spuriously associated with sex in cohorts such as the UK Biobank, likely due to their effect on differential participation 13.

10. Results: “Analysis of 116 mitochondrial genetic variants (MAF>0.01) available in the UK Biobank directly genotyped microarray data identified two associated with EWGSOP-defined low hand grip strength ($p<0.00043$, i.e. Bonferroni-adjustment for mitochondrial variants). rs41518645 is a missense variant (p.Asp171Asn) in MT-CYB, identified in Plink logistic regression analysis ($p=0.0003$). rs201950015 is intronic, located between genes CO1 and ATP6/8 ($p=0.00042$). No variants achieved “genome-wide” significant ($p<5*10^{-8}$). See Supplementary Table 5.” I suggest you remove the mitochondrial analysis from the paper. There is no replication or meta-analysis, and it adds nothing to the story. The poor methodological quality detracts from the rest of the paper.

We respectfully contest that this null finding of no genome-wide significant mtDNA variants (in UKB alone) should be removed. Mitochondria and their dysfunction are key areas of research in ageing and frailty, especially with regards to muscle function and metabolism. We were not able to perform a meta-analysis of these variants, but the analysis in UK Biobank alone we believe is worth including, at least to highlight the need of further research. We have modified the section to hopefully moderate the conclusions and allay your concerns. It now reads:

*In the analysis of 116 mitochondrial genetic variants (MAF>0.01) available in the UK Biobank directly genotyped microarray data, no variants reached “genome-wide” significance ($p>5*10^{-8}$). Two were associated with EWGSOP-defined low hand grip strength at nominal significance ($p<0.00043$, i.e. Bonferroni-adjustment for mitochondrial variants). rs41518645 is a missense variant (p.Asp171Asn) in MT-CYB, identified in Plink logistic regression analysis ($p=0.0003$). rs201950015 is intronic, located between genes CO1 and ATP6/8 ($p=0.00042$). These findings need further scrutiny in studies assessing the influence of mitochondrial dysfunction on muscle function and metabolism. See Supplementary Table 8.*

11. Results section: Subset of overall grip strength genetic variants are associated with low grip strength – this is not really results, more discussion of overlap between studies. Suggest you report in point 3 above, and move this to discussion.

We have moved this paragraph to the discussion, and reworded it to start:

*In comparison to a previous study that analysed grip strength as a continuous measure in the UK Biobank cohort 8 that included all ages (40 to 70 years), we found that only three of the 64 identified variants were significantly ($p<5*10^{-8}$) associated with EWGSOP low grip strength...*

12. Results: “although genes expressed in the tibial nerve were borderline enriched (Benjamini-Hochberg-adjusted p-value 0.1).” – remove this. Borderline is not significant. See <https://royalsocietypublishing.org/doi/10.1098/rsos.140216> for an elegant description of why.

We have removed this last part of the sentence. It now reads:

We found no significant enrichment for the genetic determinants of low grip strength in expression profiles and epigenetic changes after adjustment for multiple testing (Benjamini-Hochberg-adjusted p-value > 0.05).

13. Results: “To explore the effect of genetic predisposition to low grip strength at older ages we created an unweighted genetic risk score (GRS) in the UK Biobank European sample” Why did you use an unweighted score, rather than the more usual weighted GRS?

Use of weights from the discovery sample can bias associations and lead to overestimated effects (“Winner’s curse”). We therefore opted to use an unweighted score. We have added this sentence to that paragraph:

We opted to use an unweighted score as use of weights from analyses including the discovery sample can bias associations and lead to overestimated effects (so-called “winner’s curse”)²⁰.

20. Burgess, S. & Thompson, S. G. Use of allele scores as instrumental variables for Mendelian randomization. Int. J. Epidemiol. 42, 1134–1144 (2013).

14. Results: “co-morbidities we performed regression analyses in the UK Biobank cohort with adjustment for the following covariates: height, weight, skeletal muscle mass (determined using bioimpedance analysis), osteoarthritis, Rheumatoid arthritis, osteoporosis, Dupuytren’s contracture (one or more fingers permanently bent), and rhizarthrosis (arthritis of the thumb).” How did you determine if a person was affected by OA, RA, OP, DD, and thumb arthritis? Was this just on ICD10 codes, or did you use OPCS codes too (for example for hip or knee replacement, DD surgery etc.) and self-report? If you only used ICD-10 you will have missed a significant proportion of cases, and will need to re-run this analysis. Please also provide a supplementary table of the exact codes you have used for this analysis.

We used data from the UK Biobank baseline assessment (self-report followed up by validation of medical conditions by trained nurse) and hospital episode statistics (ICD10) data. We did not use the OPCS data to additionally identify conditions based on operations and interventions. Having considered it, many such operations and interventions (e.g. hip or knee replacement) whilst predominantly for OA/RA will not be specific to any particular disease/condition in this case (e.g. traumatic injuries, congenital abnormalities), and we hope that not including these is acceptable. We have added the details for the specific ICD10 codes used as a Supplementary Table (19)

15. Discussion: “In this study of 256,523 Europeans aged 60 years and over we identified 15 genetic loci - 3 novel - associated with the EWGSOP definition of low grip strength (dynapenia),” I thought 12 were novel?

We apologise for the confusion, which we have attempted to clarify throughout the manuscript. 12 of the 15 loci did not come up in previous analysis of continuous strength, however most have come up in GWAS of other relevant traits such as osteoarthritis. So only 2 were “novel” in that they appear for the first time in GWAS (as in the late Table 2, what is now Supplementary Table 3). We have changed that discussion section to read:

In this study of 256,523 Europeans aged 60 years and over we found that 15 genetic loci were associated with the EWGSOP definition of low grip strength (dynapenia), plus two

additional loci for the FNIH definition that used a more extreme definition for muscle weakness. Only three of these are known to be associated with continuous strength measures in GWAS, and only 3 of the 64 known overall strength loci are associated with clinically low grip here, suggesting that the genetic causes of clinically meaningful weakness at older ages are partly distinct. Two of the low grip strength-associated genetic signals identified have not appeared in GWAS prior to the time of analysis, further demonstrating that low strength in older people may have distinct genetic underpinnings...

16. Discussion: "Only three of these were previously implicated in previous analyses of continuous strength measures, suggesting that the genetic causes of clinically meaningful weakness at older ages are partly distinct." Again, add in the converse to emphasise the lack of meaningful overlap (see point 3).

We have changed the sentence to read:

Only three of these are known to be associated with continuous strength measures in GWAS, and only 3 of the 64 known overall strength loci are associated with clinically low grip here, suggesting that the genetic causes of clinically meaningful weakness at older ages are partly distinct.

17. Discussion: "of age-related diseases HLA-DQA1 was confirmed" confirmed is not the right word here, perhaps associated or identified?

We agree. Changed to "identified"

18. Discussion: "Overall five of the fifteen genomic risk loci for EWGSOP low grip strength have been previously associated, or in linkage disequilibrium with variants associated with osteoarthritis" Did you miss out the ALDH1A2 locus here?

Yes, thank you for pointing this out. We have changed the sentence to read:

Overall six of the fifteen genomic risk loci for EWGSOP low grip strength have been previously associated (or are in LD) with osteoarthritis (rs143384 – GDF5, rs13107325 – SLC39A8, rs34464763 – C12orf60, rs2899611 - ALDH1A2, rs958685 – TGFA and rs79723785 - BRSK1)...

19. Discussion: "adiposity, a known risk factor for arthritis 28." Please use osteoarthritis, not generic arthritis, which may include Rheumatoid.

Added "osteo" prefix to arthritis.

20. Discussion: "The lack of this gene in mice is perinatally lethal, however at emb" Change to knockout.

Changed to "Knocking out this gene in mice is perinatally lethal..."

21. Discussion: "Mitochondrial dysfunction is a hallmark of aging, yet we found no variants...." Suggest removal of this paragraph.

Again we respectfully ask that the paragraph be included, as mitochondria are such an important topic to study in ageing and frailty. We have modified the paragraph to highlight that this was a sub-analysis rather than full meta-analysis. The paragraph now reads:

Mitochondrial dysfunction is a hallmark of aging, yet we found no variants associated with low strength at genome-wide significance levels in a sub-analysis of UK Biobank only. Variants in MT-CYB were nominally significant (p=0.0003)...

22. Discussion: "Although puberty timing is highly polygenic, it is strongly genetically correlated with BMI" Later menarche causes taller height (which accounts for at least some of the association with lower BMI). This is a complex inter-related area, and this should be brought out in the discussion.

We have expanded the section to now read:

Although puberty timing is highly polygenic, it is strongly genetically correlated with BMI (-35%)⁵², with complex interactions: being thinner in childhood is associated with delayed menarche⁵³, but later menarche results in taller adult height⁵⁴.

23. Discussion: "we future analyses should include this." Not correct English.

We removed the word "we" from that part of the sentence.

24. Table 1. Please add whether the reported SNP is genotyped or imputed. If imputed, please add the info score. I am concerned that the info score cut off was too lax – 0.9 is a better threshold, but it is OK if the associated imputed SNPs have an info score of >0.9.

No variants were directly genotyped in all cohorts, however we have added a new supplementary table (#2 in new version) that includes the full METAL meta-analysis output for the lead SNPs, and the individual cohort BETA, SE and INFO (imputation quality) information. Three of the variants were directly genotyped in UK Biobank, and the remaining 12 had imputation scores >0.9. The vast majority had INFO scores >0.8 in all cohorts. A couple of variants in the small cohorts (such as InCHIANTI ~1,000 participants) had INFO scores <0.6 which are unlikely to have impacted the meta-analysis result.

25. Table 1. Please add the effect allele frequency.

Please see the column "EAF" (Effect Allele Frequency) in Table 1

26. Table 1. The headers and rsID numbers are in bold. As are the p-values below 5×10^{-9} . I don't find this useful, they are not special. I would prefer the entire row of the novel variants to be highlighted in bold.

We have reformatted the table as requested (the only bold rows are those that did not appear in the GWAS catalogue of published studies at the time of analysis).

27. Table 2. Suggest this goes in Supplementary.

We have made this a Supplementary Table.

Reviewer #2 (Remarks to the Author):

This is a very well written manuscript of a GWAS meta-analysis of muscle weakness that employs a large number of cases and controls aged 60 years and over. Several novel loci have been identified, with subsequent comparisons with other age-related conditions and with anthropometric traits

highlighting examples of genetic overlap. There are also the standard pathway analyses, searches for eQTL effects, MR analysis, etc. One of the striking genetic correlations was with OA. This is intriguing and the authors are commended for noting that a proportion of this may be the impact that painful hand OA will have on the ability to adequately undertake a grip strength test.

Overall, I think there are some novel and interesting insights reported, the investigators have undertaken all necessary analyses, and they have drawn reasonable and logical conclusions.

I do not have any changes to recommend.

Thank you for the positive comments and feedback! We have acknowledged the possibility of hand OA affecting grip strength in our response to reviewer 1 comment 4.

REVIEWER COMMENTS

Reviewer #1 (Remarks to the Author):

Thank-you for your responses to my original comments. All issues have been dealt with except for point 14:

Original comment

Results: "co-morbidities we performed regression analyses in the UK Biobank cohort with adjustment for the following covariates: height, weight, skeletal muscle mass (determined using bioimpedance analysis), osteoarthritis, Rheumatoid arthritis, osteoporosis, Dupuytren's contracture (one or more fingers permanently bent), and rhizarthrosis (arthritis of the thumb)." How did you determine if a person was affected by OA, RA, OP, DD, and thumb arthritis? Was this just on ICD10 codes, or did you use OPCS codes too (for example for hip or knee replacement, DD surgery etc.) and self-report? If you only used ICD-10 you will have missed a significant proportion of cases, and will need to re-run this analysis. Please also provide a supplementary table of the exact codes you have used for this analysis.

Response

We used data from the UK Biobank baseline assessment (self-report followed up by validation of medical conditions by trained nurse) and hospital episode statistics (ICD10) data. We did not use the OPCS data to additionally identify conditions based on operations and interventions. Having considered it, many such operations and interventions (e.g. hip or knee replacement) whilst predominantly for OA/RA will not be specific to any particular disease/condition in this case (e.g. traumatic injuries, congenital abnormalities), and we hope that not including these is acceptable. We have added the details for the specific ICD10 codes used as a Supplementary Table (19)

Further comment

I still think that OPCS and self-reported operation are very important, specifically for OA and Dupuytren Disease. For example:

<https://www.biorxiv.org/content/10.1101/2020.05.14.095653v1.supplementary-material>

See supplementary table 1 of this pre-print. Here we were able to classify a further c11,000 participants as cases rather than controls compared to a previous analysis of UK Biobank, by utilising surgical codes. This avoids misclassification bias. For some of these diagnoses (eg Dupuytren Disease), there are very well defined OPCS and self-report codes that identify surgical cases (see below). Furthermore, because surgery has inherent risks of complications, it is usually reserved for the phenotypically more severe cases, which may well also be genotypically "more severe", so misclassifying these cases as controls in a genetic analysis is especially bad.

The argument about the underlying "cause" of a hip or knee replacement (for example) doesn't really hold. The vast majority of these operations are performed for OA (and you could simply exclude those participants with one of your RA codes to clean the dataset). Furthermore, there is a vast literature on the "cause" of OA, with most people favouring a combination of mechanical loading ("trauma") and genetic predisposition (Which may well act .

I still think this should be re-analysed. Ultimately, that decision will depend on the Journal Editor. Otherwise, this paper is excellent and ready for publication.

Reviewer #1 (Remarks to the Author):

Thank-you for your responses to my original comments. All issues have been dealt with except for point 14:

Original comment

Results: “co-morbidities we performed regression analyses in the UK Biobank cohort with adjustment for the following covariates: height, weight, skeletal muscle mass (determined using bioimpedance analysis), osteoarthritis, Rheumatoid arthritis, osteoporosis, Dupuytren's contracture (one or more fingers permanently bent), and rhizarthrosis (arthritis of the thumb).” How did you determine if a person was affected by OA, RA, OP, DD, and thumb arthritis? Was this just on ICD10 codes, or did you use OPCS codes too (for example for hip or knee replacement, DD surgery etc.) and self-report? If you only used ICD-10 you will have missed a significant proportion of cases, and will need to re-run this analysis. Please also provide a supplementary table of the exact codes you have used for this analysis.

Response

We used data from the UK Biobank baseline assessment (self-report followed up by validation of medical conditions by trained nurse) and hospital episode statistics (ICD10) data. We did not use the OPCS data to additionally identify conditions based on operations and interventions. Having considered it, many such operations and interventions (e.g. hip or knee replacement) whilst predominantly for OA/RA will not be specific to any particular disease/condition in this case (e.g. traumatic injuries, congenital abnormalities), and we hope that not including these is acceptable. We have added the details for the specific ICD10 codes used as a Supplementary Table (19)

Further comment

I still think that OPCS and self-reported operation are very important, specifically for OA and Dupuytren Disease. For example:

<https://www.biorxiv.org/content/10.1101/2020.05.14.095653v1.supplementary-material>

See supplementary table 1 of this pre-print. Here we were able to classify a further c11,000 participants as cases rather than controls compared to a previous analysis of UK Biobank, by utilising surgical codes. This avoids misclassification bias. For some of these diagnoses (eg Dupuytren Disease), there are very well defined OPCS and self-report codes that identify surgical cases (see below). Furthermore, because surgery has inherent risks of complications, it is usually reserved for the phenotypically more severe cases, which may well also be genotypically “more severe”, so misclassifying these cases as controls in a genetic analysis is especially bad.

The argument about the underlying “cause” of a hip or knee replacement (for example) doesn't really hold. The vast majority of these operations are performed for OA (and you could simply exclude those participants with one of your RA codes to clean the dataset). Furthermore, there is a

vast literature on the “cause” of OA, with most people favouring a combination of mechanical loading (“trauma”) and genetic predisposition (Which may well act .

I still think this should be re-analysed. Ultimately, that decision will depend on the Journal Editor. Otherwise, this paper is excellent and ready for publication.

I have attached a table of DD codes and link to the most definitive OA GWAS to help with this.

Response 16th Sept 2020

We thank the reviewer for the comments and attached table of DD codes, which we have used to identify additional cases based on OPCS code. To confirm, we have repeated the sensitivity analysis of the SNP~low grip associations with adjustment for osteoarthritis or Dupuytren’s contracture where we originally included self-reported or hospital diagnosed codes only to now include OPCS codes as requested (hip or knee replacements for OA, and the attached codes for Dupuytren’s). The number of cases of osteoarthritis in the analysis sample has increased from 27,653 to 29,380 following inclusion of hip or knee replacement codes. The number of Dupuytren’s contracture cases has increased from 1,409 to 1,455 in the analysis sample (in 60+ year olds of genetically European ancestry without missing data for grip strength and covariates).

We have made the following changes to the manuscript and accompanying files:

Manuscript file, Results section for “Low grip strength loci independence from musculoskeletal traits and diseases”

*To determine whether the genetic variants associated with low grip strength identified in the GWAS were independent of anthropometric traits or **prevalent** musculoskeletal co-morbidities we performed regression analyses in the UK Biobank cohort with adjustment for the following covariates: height, weight, skeletal muscle mass (determined using bioimpedance analysis), osteoarthritis, Rheumatoid arthritis, osteoporosis, Dupuytren’s contracture (one or more fingers permanently bent), and rhizarthrosis (arthritis of the thumb). **Disease diagnoses were either self-reported, hospital diagnosed, or inferred from relevant surgical procedures (for example Palmar Fasciectomy to treat Dupuytren’s contracture, and hip or knee replacements resulting from osteoarthritis). See Supplementary Table 19 for diagnostic and surgical codes used.***

The association between eight of the fifteen EWGSOP loci and low grip strength was attenuated after adjusting for height, including rs143384 (initial UKB $p=3.7 \times 10^{-11}$; adjusted UKB $p=3.8 \times 10^{-2}$) and rs7624084 (initial UKB $p=9.3 \times 10^{-7}$; adjusted UKB $p=4.9 \times 10^{-1}$). Adjustment for weight or BMI did not substantially attenuate any of the associations. Overall, the associations were not attenuated by adjustment for osteoarthritis, Rheumatoid arthritis, osteoporosis, Dupuytren’s, or rhizarthrosis. See Supplementary Table 20 for detailed results.

Supplementary Table 19 now also includes the OPCS codes used, and also include a “N” column to show the number of participants in the analysis with a diagnosis for each condition:

Condition	UK Biobank self-reported fields	ICD-10 codes	OPCS codes	N
Osteoarthritis	1465	M15.0; M15.1; M15.2; M15.9; M16.0; M16.1; M17.0; M17.1; M18.0; M18.1; M19.0	O18*; W40*; W41*; W42*; W37*; W38*; W39*; W46*; W47*; W48*; W93*; W94*; W95*	29,380
Rheumatoid arthritis	1464	M05; M06		3,263
Rhizarthrosis		M18; M180; M181; M182; M183; M184; M185; M189		355
Osteoporosis	1309	M80; M81		6,308
Dupuytren's contracture	1544	M720; M7204	T521; T522; T525; T526; T541; T561; T562	1,455
Any autoimmune condition	1234; 1522; 1428; 1464; 1381; 1313; 1382; 1261; 1456; 1463; 1477; 1453; 11222	D69*; D758; D141; D862; E271; E05; E051; E063; E100; E101; E102; E103; E104; E105; E106; E107; E108; E109; G35; G737; G70*; H30*; K900; K510; K512; K513; K514; K515; K518; K519; L40; L400; L401; L402; L403; L404; L405; L408; L409; L12*; L10*; L511; M023; M0230; M0236; M0239; M028; M0281; M0284; M0285; M0286; M0287; M029; M0290; M0293; M0294; M0295; M0296; M0297; M0299; M06*; M320; M321; M328; M329; M45; M350; M30*; M34*; M352		16,951

Supplementary Table 20 shows the updated results from the models in UK Biobank adjusted for prevalent diagnoses, including operations codes for OA (hip or knee replacement) and Dupuytren's contracture (the codes provided by the reviewer). The estimates have changed slightly but the overall conclusion that the associations between the variants and low grip strength are consistent after adjustment for relevant musculoskeletal conditions remains the same:

SNP	Effect allele	Adjustment	OR	95% CIs		P
rs12140813	T	+ osteoarthritis	1.060	1.037	1.083	1.8E-07
rs12140813	T	+ Dupuytren's contracture	1.060	1.037	1.084	1.3E-07
rs958685	C	+ osteoarthritis	1.045	1.027	1.063	7.6E-07
rs958685	C	+ Dupuytren's contracture	1.048	1.030	1.066	8.8E-08
rs7624084	T	+ osteoarthritis	1.044	1.026	1.063	1.1E-06
rs7624084	T	+ Dupuytren's contracture	1.045	1.027	1.063	9.1E-07
rs13107325	T	+ osteoarthritis	1.099	1.065	1.135	7.0E-09
rs13107325	T	+ Dupuytren's contracture	1.108	1.073	1.144	3.1E-10
rs34415150	G	+ osteoarthritis	1.094	1.070	1.118	2.1E-15
rs34415150	G	+ Dupuytren's contracture	1.093	1.070	1.118	1.9E-15
rs10952289	T	+ osteoarthritis	1.042	1.023	1.061	1.3E-05
rs10952289	T	+ Dupuytren's contracture	1.044	1.025	1.064	3.2E-06
rs11236213	G	+ osteoarthritis	1.061	1.042	1.082	5.2E-10
rs11236213	G	+ Dupuytren's contracture	1.062	1.042	1.082	4.1E-10
rs34464763	A	+ osteoarthritis	1.059	1.041	1.078	1.6E-10
rs34464763	A	+ Dupuytren's contracture	1.060	1.042	1.079	7.0E-11
rs3118903	A	+ osteoarthritis	1.069	1.047	1.091	2.2E-10
rs3118903	A	+ Dupuytren's contracture	1.067	1.045	1.089	7.2E-10
rs2899611	G	+ osteoarthritis	1.048	1.030	1.066	1.3E-07
rs2899611	G	+ Dupuytren's contracture	1.048	1.030	1.066	9.6E-08
rs143459567	T	+ osteoarthritis	1.140	1.092	1.191	2.8E-09
rs143459567	T	+ Dupuytren's contracture	1.137	1.089	1.188	4.8E-09
rs8061064	A	+ osteoarthritis	1.039	1.021	1.057	1.7E-05
rs8061064	A	+ Dupuytren's contracture	1.038	1.021	1.057	1.9E-05
rs62102286	T	+ osteoarthritis	0.953	0.936	0.970	5.6E-08
rs62102286	T	+ Dupuytren's contracture	0.950	0.934	0.967	8.4E-09
rs79723785	C	+ osteoarthritis	0.852	0.796	0.912	3.5E-06
rs79723785	C	+ Dupuytren's contracture	0.846	0.791	0.905	1.1E-06
rs143384	A	+ osteoarthritis	0.943	0.927	0.960	8.1E-11
rs143384	A	+ Dupuytren's contracture	0.942	0.926	0.959	3.5E-11